# Analyzing Film Adaptation through Narrative Alignment

**Tanzir Pial, Shahreen Salim, Charuta Pethe, Allen Kim, Steven Skiena**
Department of Computer Science,
Stony Brook University, NY, USA
{tpial,ssalimaunti,cpethe, allekim, skiena}@cs.stonybrook.edu

## Abstract

Novels are often adapted into feature films, but the differences between the two media usually require dropping sections of the source text from the movie script. Here we study this screen adaptation process by constructing narrative alignments using the Smith-Waterman local alignment algorithm coupled with SBERT embedding distance to quantify text similarity between scenes and book units. We use these alignments to perform an automated analysis of 40 adaptations, revealing insights into the screenwriting process concerning (i) faithfulness of adaptation, (ii) importance of dialog, (iii) preservation of narrative order, and (iv) gender representation issues reflective of the Bechdel test.

## 1 Introduction

Book-to-film adaptation plays an important role in film-making. From an artistic point of view, there are many possible ways to craft a movie from any given book, all of which entail changes (addition, deletion, and modification) to the structure of the text. Books and films are different forms of media, with different characteristics and goals. Novels are generally much longer texts than screenplays, written without the strict time-limits of film. Further, film adaptations present the screenplay writers' interpretation of the book, whose intentions are often different from the author of the original text.

In this paper, we study which parts of books get excised from the script in their movie adaptations. Our technical focus is on methods to identify alignments between each unit (e.g., dialog, set-up of a scene) of a movie script with the paragraphs of the book. However, certain book paragraphs must be left unmatched when appropriate, reflecting sections of the book deleted in the movie adaptation for simplifying or focusing the action. We primarily focus on understanding screenwriter behavior and discovering patterns in the book-to-film adaptation process.

The importance of book-to-film adaptation is evidenced by its economic and cultural impact. Frontier Economics (2018) found that books were the basis of more than 50% of the top 20 UK-produced films between 2007-2016. These films grossed £22.5bn globally, accounting for 65% of the global box office gross of the top UK films. Recent availability of large-scale book and screenplay data empowers the NLP community to study the screenwriting process. Our work opens new lines of research in this domain, providing insight into the thought processes of screenwriters and film producers.

The relationship between book source and film script has repeatedly been the subject of research within the NLP community. The degree to which movie adaptations are faithful to the original books was raised by Harold (2018), who considers two questions: how it can be decided if a movie is faithful to its book, and whether being faithful to its source is a merit for a movie adaptation. There have been many works focusing on analyzing movie adaptations of specific books, such as Rahmawati et al. (2013) and Österberg (2018).

With the advent of GPT-3 (Brown et al., 2020) and more advanced models, Artificial Intelligence (AI) systems are increasingly being proposed to automate creative tasks in the entertainment industry. The Writers Guild of America's historic strike highlighted concerns about AI's use in covered projects – specifically, there was a contention that AI should not be involved in the direct writing or rewriting of literary material, nor should it be utilized as source material (WGA, 2023). A book to film script alignment and analysis tool has potential application for both sides of the debate, e.g., by creating training dataset for generative models for rewriting creative materials into different formats and analyzing the differences between materials created by humans

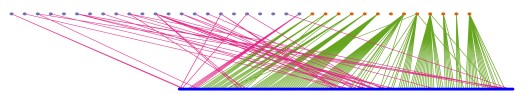

| (a) Movie: HP7 Part 1 | (b) Movie: HP7 Part 2 |

Figure 1: When one book becomes two movies. Alignments of the seventh Harry Potter book (top line) with scripts for the Part 1 (left) and Part 2 (right) scripts. The green edges denote alignment with the correct portion of the book, as opposed to the magenta edges. Our local alignment methods recognize the split structure of the book much better than global alignment methods like *Book2Movie* (Tapaswi et al., 2015), who align only 41.4% of Part 2 with the second part of the book, compared to 81.6% with our alignment.

and AI, as well as identifying areas where AI may have limitations.

We frame the book-to-script problem as a sequence alignment problem, where a book and its movie script are two text sequences. Sequence alignment techniques are critically important in bioinformatics, specifically for efficiently aligning genome and protein sequences. In the field of computer vision, some of these algorithms have been adapted for image matching (Thevenon et al., 2012). Pial and Skiena (2023) posit that these algorithms have a role to play in the analysis of long texts in NLP too. We adapt their proposed text alignment methods for the book-to-film adaptation domain. The primary mechanism of most sequence alignment algorithms does not change based on the components of the sequence. Provided with a definition for similarity (or distance) between individual components, they can be adapted to any domain.

Our primary contributions[1] include:

- *Book-Script Alignment Algorithm* – To identify commonalities between a book and its script adaptation, we introduce a domain-adapted book-to-film alignment approach. This method builds upon the techniques advanced by Pial and Skiena (2023) for general narrative alignment. It leverages modern text embedding methodologies in conjunction with local alignments generated by the Smith-Waterman (SW) algorithm (Smith and Waterman, 1981), a seminal technique in computational biology. Our approach proves significantly more accurate than prior methods including *Book2Movie* (Tapaswi et al., 2015). Our alignments identify which portions of the book are excised in the adaptation, providing novel insights into the screenwriting process.

- *Recognizing Faithfulness of Adaptations* – Certain film adaptations retain only the title of the putative source book, while others strive to fully capture the contents and spirit of the source. We use our alignment methods to define a faithfulness statistic (retention percentage), and demonstrate its performance on 18 book-script pairs where published critics have commented on the faithfulness of the adaptation. We achieve a Spearman rank correlation of 0.65 between retention percentage and critic labels, significant to the 0.001 level.

- *Investigations of Screenwriter Behavior* – For 40 book-film adaptations, we analyze screenwriter decisions from several perspectives:

  - We demonstrate that book-units with dialog are substantially more likely to be retained than other narrative text, significant with a p-value $< 3 \times 10^{-6}$.

  - We prove that the narrative order of book-script pairs are highly consistent, demonstrating that local Smith-Waterman alignments capture global narrative order.

  - We show that script alignments can reveal whether the adaptation is likely to pass the gender-role sensitivity Bechdel test based on text from a given book.

- *Datasets for Text-to-Text Alignment* – We introduce three novel datasets designed to facilitate research on the book-to-movie alignment task. The first consists of six book-script pairs where each unit of the book is annotated as either retained or removed in the corresponding screenplay. The second dataset contains two book-script pairs where we fully align individual chapters of the book with the corresponding scenes in the movie. Finally, we provide a dataset of 40 book-script pairs to ex-

---

[1]All codes and datasets are available at https://github.com/tanzir5/alignment_tool2.0

plore additional research questions associated with film adaptation. Our software, testing procedures, and data resources will be made publicly available upon acceptance of this paper, subject to copyright limitations on the original data.

The outline of this paper is as follows. We survey work on script analysis and alignment methods in Section 2. Our alignment methods based on Smith-Waterman and SBERT embeddings are presented in Section 3, and the datasets we develop in Section 4. The accuracy of our alignment algorithms is established in Section 5, before we apply these methods to four aspects of film adaptation in Section 6.

## 2 Related Work

The task of aligning the video of a movie with a book has been explored in prior research. Tapaswi et al. (2015) proposed a graph-based approach to align book chapters with corresponding movie scenes, treating the alignment as a shortest path finding problem. Zhu et al. (2015) utilized neural sentence embeddings and video-text neural embeddings to compute similarities between movie clips and book sentences. In contrast, our work primarily analyzes the removed and retained parts of the book to gain insights into screenwriter behavior while also performing book-to-film alignment.

Core to our proposed method is the dynamic programming (DP) algorithm for alignment. DP and the associated dynamic time warping algorithms have been widely applied in various film alignment tasks, including aligning script dialogs with transcript dialogs (Everingham et al., 2006; Park et al., 2010), aligning scripts with video for person identification (Bauml et al., 2013), action recognition (Laptev et al., 2008), and identifying scene order changes in post-production (Lambert et al., 2013), aligning plot synopses with video (Tapaswi et al., 2014), and annotating movie videos with script descriptions (Liang et al., 2012). Kim et al. (2020) uses DP to segment novels into parts corresponding to a particular time-of-day.

Neural machine translation (Bahdanau et al., 2014) models create soft alignments between target words and segments of the source text relevant for predicting the target word. The size of the book and film scripts restrict us to directly employ neural models like this. Other than translations involving two languages, Paun (2021) proposes using

| Method | HP2 | HP4 | Complexity |
|---|---|---|---|
| Baseline (Length) | 22.6 | 3.0 | $O(c+s)$ |
| *Book2Movie* | 59.5 | 52.5 | $O(c^2 s \log(cs))$ |
| Greedy + SBERT | 61.2 | 51.5 | $O(mn \log(mn))$ |
| SW + SBERT | **85.3** | **66.2** | $O(mn \log(mn))$ |

Table 1: Comparison of alignment accuracy and efficiency of different algorithms. For Baseline (Length) we align each chapter sequentially with a number of scenes one by one according to the length of the chapters. $s$ = no. of scenes, $c$ = no. of chapters, $m$ = no. of book paragraphs, $n$ = no. of scene paragraphs.

Doc2Vec (Le and Mikolov, 2014) embeddings for creating monolingual parallel corpus composed of the works of English early modern philosophers and their corresponding simplified versions.

Additionally, several previous works have addressed different aspects of film scripts, such as genre classification (Shahin and Krzyżak, 2020), estimating content ratings based on the language use (Martinez et al., 2020), script summarization (Gorinski and Lapata, 2015), sentiment analysis (Frangidis et al., 2020), automated audio description generation (Campos et al., 2020), creating dialog corpora (Nio et al., 2014), creating multimodal treebanks (Yaari et al., 2022). Lison and Meena (2016) performs subtitle to movie script alignment for creating training dataset for automatic turn segmentation classifier. However, to the best of our knowledge, ours is the first-ever work on automated analysis of book-to-film adaptations on a moderately large scale (40 books).

In bioinformatics, sequence alignment has been studied in depth for many decades. Accurate but comparatively slower classes of algorithms include the DP algorithms like local alignments by Smith and Waterman (1981), global alignments by Needleman and Wunsch (1970), etc. Global alignments align the whole sequence using DP, while local alignments try to align locally similar regions. In this work, we use Smith-Waterman (SW) to produce local alignments, which are subsequently used to construct the final global alignment. Heuristic but faster algorithms include Altschul et al. (1990).

## 3 Aligning Books with Film Scripts

Here we describe the Smith-Waterman algorithm for local sequence alignment, including the similarity metrics we use to compare books and scripts.

## 3.1 The Smith-Waterman Algorithm

We modify the classical Smith-Waterman algorithm (SW) to allow for many-to-many matching, since a paragraph in the book can be conceptually aligned with multiple units of the script. We calculate the highest alignment score for two sequences $X$ and $Y$ of length $m$ and $n$, respectively, using the recurrence relation $H(i, j)$.

$$H(i,j) = \max \begin{cases} H(i-1, j-1) + S(X_i, Y_j), \\ H(i-1, j) + g + \max(0, S(X_i, Y_j)), \\ H(i, j-1) + g + \max(0, S(X_i, Y_j)), \\ 0 \end{cases}$$

$$(1)$$

To identify the most similar segment pair, the highest-scoring cell in the SW DP matrix $H$ is located, and segments are extended left towards the front of the sequences until a zero-score cell is reached. The contiguous sub-sequence from the start cell to the zero-score cell in each sequence forms the most similar segment pair.

The algorithm allows for matching two components using a similarity metric or deleting a component with a gap penalty, $g$. We set $g = -0.7$ empirically. The algorithm has a time complexity of $O(mn)$.

## 3.2 Independent Matches in Local Alignments

A complete book-script alignment consists of a collection of local alignments, but there is no single standard strategy in the literature for creating global collections from local alignment scores. Here we use a greedy approach, processing matching segments in decreasing alignment score order. To eliminate weak and duplicate local alignments, we skip over previously-selected paragraph pairs. The cost of sorting the DP matrix scores results in an algorithm with $O(mn \log(mn))$ time complexity.

Using the paragraph-level alignments produced by SW, we compute information at two different levels of granularity: decision on the removal of book units (usually made up of 5-10 paragraphs, as discussed in Section 4.1) and alignment of chapters (made up of 25-50 paragraphs). Generally, book units prove a more suitable choice. Conversely, aligning chapters with scenes provides a universally recognized evaluation method employed in previous works (Tapaswi et al., 2015).

We interpret the removal of a book unit from the script whenever less than half of its paragraphs align with any scene paragraph. In case of chapter alignment, each scene paragraph with an alignment casts a vote for the chapter of the aligned book paragraph. We align the chapter with the maximum number of votes with the scene.

## 3.3 Similarity Metric

We use paragraphs as the smallest unit for alignment and generate semantically meaningful embeddings for each book and script paragraph using the pretrained SBERT model (Reimers and Gurevych, 2019). While SBERT is primarily designed for sentence embeddings, it has been trained on diverse datasets for various purposes. We use the SBERT model[2] pretrained on the MS MARCO (Nguyen et al., 2016) dataset for information retrieval. This particular model is trained to maximize cosine similarity between relevant query and passage pairs. Given that script paragraphs typically consist of one to two sentences resembling single-sentence queries in the training data, and book paragraphs are generally longer akin to multi-sentence passages in the training data, this model is well-suited for our method.

Dayhoff et al. (1978) suggests that DP based algorithms perform well when the expected similarity score of two random unrelated components is negative and only related components have a positive similarity score. To incorporate this idea, we compute the similarity score between two paragraphs $x$ and $y$ as,

$$S(x, y) = \sigma(Z(x, y)) - th_s) \times 2 - 1 \quad (2)$$

where $\sigma(.)$ denotes the logistic sigmoid function, $Z(., .)$ denotes the z-score of the cosine similarity between $x$ and $y$ derived from a distribution of similarity scores between random paragraphs, and $th_s$ denotes a threshold for having a positive score. The threshold $th_s$ ensures that all pairs with less than $th_s$ z-score have a negative similarity score. We set $th_s$ to +0.6 empirically, but found that all thresholds from +0.5 to +1 z-score performed similarly.

## 4 Datasets

This study focuses on three data sets of annotated book-script pairs:

- **Removed Parts Annotation Set.** For six book-script pairs (four Harry Potter and two Jane Austen classics), we manually annotated

---

[2] https://www.sbert.net/docs/pretrained-models/msmarco-v3.html

| Alignment Method | Similarity Metric | PB | HP2 | HP4 | HP7P1 | HP7P2 | PP |
|---|---|---|---|---|---|---|---|
| Greedy (Baseline) | jaccard | 0.68 | 0.64 | 0.32 | 0.34 | 0.38 | 0.67 |
| | glove | 0.57 | 0.58 | 0.27 | 0.28 | 0.31 | 0.55 |
| | hamming | 0.54 | 0.51 | 0.29 | 0.12 | 0.35 | 0.52 |
| | tf-idf | 0.63 | 0.63 | 0.29 | 0.32 | 0.32 | 0.64 |
| | SBERT | **0.69** | 0.72 | 0.43 | 0.40 | 0.45 | 0.64 |
| Smith-Waterman (SW) | jaccard | 0.60 | 0.72 | 0.45 | 0.73 | 0.63 | 0.6 |
| | glove | 0.56 | 0.58 | 0.27 | 0.37 | 0.33 | 0.54 |
| | hamming | 0.42 | 0.48 | 0.30 | 0.24 | 0.20 | 0.47 |
| | tf-idf | 0.59 | 0.68 | 0.41 | 0.64 | 0.55 | 0.55 |
| | SBERT | 0.67 | **0.73** | **0.48** | **0.74** | **0.73** | **0.65** |

Table 2: F1 scores of different methods to determine whether book unit is retained or removed in the film adaptation. SBERT+SW achieves the highest F1 score on six of seven books. *HP* denotes the *Harry Potter* series by JK Rowling while *PB* and *PP* denote *The Princess Bride* by William Goldman and *Pride and Prejudice* by Jane Austen respectively

each book unit as *retained* or *removed*. They are listed in Table A.1 of the appendix. We note that the dataset contains five books, not six, because two movies were adapted from the seventh Harry Potter book (HP7).

- **Manual Alignment Set.** We manually aligned two books with their movie adaptation scripts. We aligned each scene with a chapter of the book. The two book-script pairs are two volumes of the *Harry Potter* series (HP2 and HP4, books also used in the first dataset).

- **Automated Analysis Set.** We collected 40 book-script pairs for automated analysis from a mix of Gutenberg (n.d.), the HathiTrust Digital Library (n.d.), and personal sources. Information about these books is provided in Table B.1 where each book-film is given a unique ID and Table B.2 in the appendix.

We collected the scripts from popular websites that publicly host scripts using the code developed by Ramakrishna et al. (2017). The length of books in pages and associated films in minutes have a Pearson correlation of 0.36, indicating that longer books get made into longer films. The process of manual annotation involved multiple researchers, but each book-movie pair was annotated by a single person without any involvement of other annotators.

## 4.1 Segmenting Texts

Film scripts generally follow a typical format with *slug lines* at the beginning of each scene that provide information such as location and time of day (Cole and Haag, 2002). This standardized format facilitates parsing and segmentation of movie scripts into individual scenes. In the case of film scripts, a paragraph usually corresponds to a single bit of dialog or set-up of the scene.

In contrast, books generally lack such a rigid structure. While chapters serve as a common unit of segmentation for books, they are often too lengthy for analyzing retention statistics. To investigate the scriptwriter's decisions regarding the retention of book parts, it becomes necessary to segment books into smaller units. We adopt the text segmentation algorithm proposed by Pethe et al. (2020). This algorithm leverages a weighted overlap cut technique to identify local minimas as breakpoints where the word usage transitions to a different subset of vocabulary indicating a context switch. These resulting text units are referred to as *book units*.

## 5 Evaluation

We evaluate the performance of our alignment algorithm against several baselines and previous work. There are two distinct aspects to be considered in our evaluation: (i) the performance of different text similarity metrics, and (ii) local vs. global alignment approaches such as *Book2Movie* (Tapaswi et al., 2015). Additionally, we provide qualitative evaluation of our alignments in Appendix C.

## 5.1 Baselines

**Different Similarity Metrics:** To test the impact of similarity metric on performance, we evaluate five different versions of SW, using the similarity metrics of Jaccard similarity, a word embedding-based metric (GloVE), Hamming distance, TF-IDF and SBERT, respectively. Details of these metrics appear in Appendix D.

**Greedy Baselines:** We formulate baseline algorithms which uses greedy matching, where the pairs of paragraphs are aligned in decreasing or-

der of their similarity values while maintaining a 1-1 book-script match. Both SBERT+SW and SBERT+greedy use SBERT embeddings as the similarity metric, but SW can utilize the similarity information of the neighborhood paragraphs during alignment, which the greedy baseline cannot. Therefore any improvement of SW+SBERT over baseline must be attributed to the SW alignment method.

## 5.2 Retention Accuracy

Book retention is a weaker standard than fine alignment. We manually annotated each book unit as retained or removed in the movie for the six book-script pairs in Table A.1. In this context, a false positive book unit denotes that the algorithm marked this unit as retained wrongly, while a true positive book unit denotes that both the algorithm and the manual annotator marked this unit as retained. We use 6,546 book units in this experiment.

Our results are presented in Table 2. The SBERT and Smith-Waterman combination produced the best F1 score for retention classification for five of the six book-script pairs. On the remaining instance it finished second to SBERT with the greedy heuristic. Based on these results, we will use SBERT+SW on our film adaptation analysis to follow.

## 5.3 Alignment Accuracy

Building gold-standard alignments between book-length novels and scripts is a challenging task. We manually aligned movie scenes with book chapters for two book-script pairs (HP2 and HP4) and compared our SW+SBERT method against *Book2Movie* (Tapaswi et al., 2015) and Greedy+SBERT baselines. Accuracy is measured as the percentage of correctly aligned scenes.

The *Book2Movie* method formulates the alignment problem as a shortest-path finding task in a graph, employing Dijkstra's algorithm to solve it. They utilize dialogue and character frequency as their similarity metric and directly align chapters with scenes. We created a simple dialog and character frequency based similarity metric to fit into the *Book2Movie* alignment algorithm. Our method's alignment accuracy is 25.8% and 13.7% more than *Book2Movie* for HP2 and HP4 respectively, as presented in the Table 1.

The *Book2Movie* algorithm assumes global priors, so that a book chapter towards the end of

the text gets a penalty for aligning with an early movie scene and vice versa. This technique aims to restrict alignments between distant scenes and chapters. However this assumption becomes troublesome when the two sequences do not follow similar order e.g., the seventh HP book and the two movies adapted from it. Our method does not assume a global prior, allowing it to successfully align both the movies with the correct parts of the book, as shown in Figure 1. *Book2Movie* correctly aligns the whole first movie with the first part of the book, while our method aligns 97.5% of the first movie correctly. But *Book2Movie* aligns only 41.4% of the second film correctly, whereas our method achieves 81.6% accuracy there. This proves how global priors can negatively influence accuracy of alignment algorithms.

## 6 Analysis of the Film Adaptation Process

The economics of the movie industry imposes stronger constraints on the length of scripts than the publishing industry forces on books. Our alignments illustrate the relationship between book and script lengths, and running times. Figure 2 (left) shows that the retention percentage of a book in the movie has an inverse relationship with the length of the book, whereas Figure 2 (middle) shows that longer movies tend to retain more parts of the books. Finally, Figure 2 (right) shows that longer books tend to have longer movie adaptations. Script retention information for all book-movie pairs in our dataset is presented in the Figure 3, using alignments constructed by our SW+SBERT method.

The book-script alignment methods we have developed so far enable us to analyze the film adaptation process in several interesting ways. The subsections below will consider four particular tasks, namely: (i) measuring the faithfulness of a film adaptation to its original source, (ii) the influence of dialog on script retention, (iii) the preservation of temporal sequencing in adaptation, and (iv) gender representation decisions in film adaptation.

## 6.1 Faithfulness of Movies to Source Novels

The faithfulness of movie adaptations to their original source material has long been a topic of research and discussion (Leitch, 2008). Our alignment methods provide to quantify the faithfulness of scripts to their source books.

We use the alignments for 40 book-movie pairs

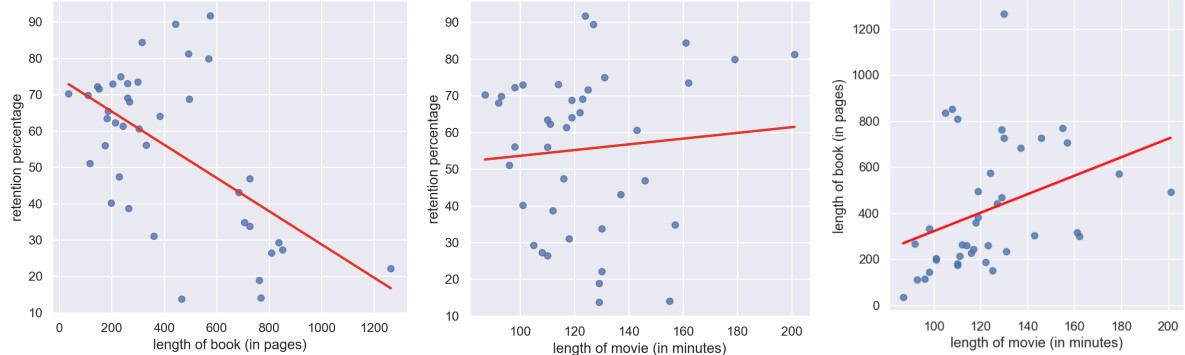

Figure 2: Relationships between book length, script length, and playing time. Scripts contain smaller fractions of longer books: correlation -0.58 (on left). Scripts for longer movies contain larger fractions of the book: correlation 0.09 (in middle). Longer books lead to longer movies: correlation 0.36 (on right).

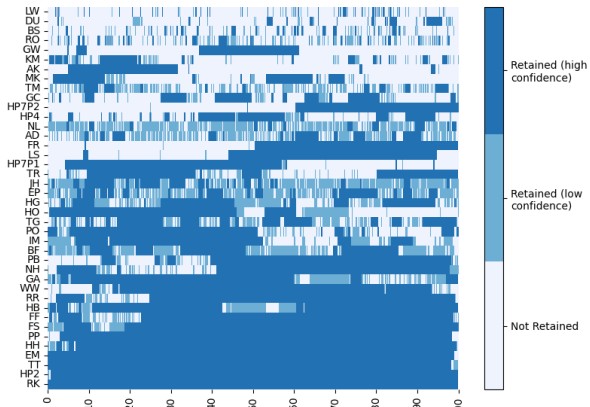

Figure 3: Alignments for 40 book-movie scripts. Here the parts of the book are colored based on if they are not retained or retained, with low or high confidence, by SW+SBERT according to the generated alignment scores.

presented in the Figure 3 to identify which text in each book gets aligned, marking it as *retained* in the script. We propose using the percentage of retained text as a metric for measuring script faithfulness, acknowledging that films completely focused on a small section of the book may be classified as unfaithful.

Table 3 presents the books with the highest and lowest amount of retention respectively. The second Jurassic Park film (The Lost World), based on the novel by Michael Crichton with the same name, had the lowest retention in our dataset, which supports claims that the movie was "*an incredibly loose adaptation of Crichton's second book*" (Weiss, 2023). On the other end of the spectrum, both critics (Blohm, 2016) and our alignments finds two movie adaptations based on Jane Austen's novels to be the most faithful to its source material.

We have annotated 18 movie adaptations with binary labels indicating faithfulness based on critic reviews. We describe the labelling process and the data sources in Appendix E. The retention percentage demonstrates a Spearman rank correlation of 0.65 with a p-value of 0.001 and achieves an AUC score of 0.875 for classifying faithfulness.This strengthens the cause for using retention percentages found by SW+SBERT as a faithfulness metric.

## 6.2 Retention of Dialog from Books

Dialogs between characters is a critical aspect of visual media. We hypothesize that movie adaptations tend to retain the more dialog-heavy parts of the book than descriptive parts without speech.

To quantify this preference, we compute a dialog retention ratio $u_b$ for each book-movie pair $b$ as:

$$u_b = \frac{|r_b \cap d_b|}{|d_b|} \times \frac{|b|}{|r_b|} \quad (3)$$

where $r_b$ denotes the retained part of book $b$, and $d_b$ denotes the part of book $b$ with dialogs. Similarly we compute $v_b$, the representative value for retaining texts without dialogs. A value greater than 1 for $u_b$ signifies that dialog-heavy parts are retained more than non-dialog parts. Figure 4 shows that for majority of the books, texts with dialog are retained more than texts without it.

Statistically, we find that dialog has mean retention of $u = 1.04$, compared to non-dialog with a mean retention of $v = 0.97$. A t-test comparing the distribution of $u_b$ and $v_b$ over all books in our data set shows that they follow two different distributions with a p-value $< 3 \times 10^{-6}$. The effect size is calcu-

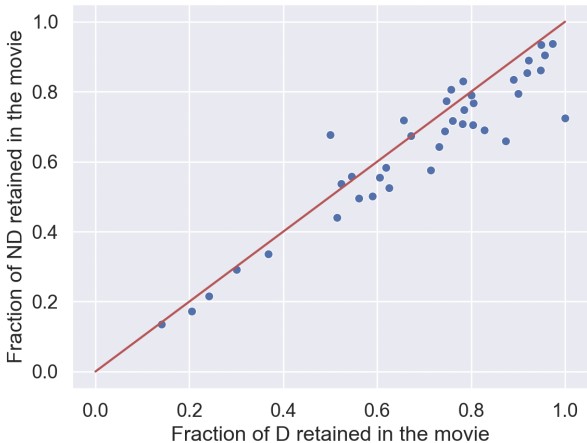

Figure 4: Paragraphs of books with dialog (D) are more likely to be retained in the script than paragraphs containing no dialog (ND). Each blue point denotes a book-script pair, represented by the fraction of dialog and no-dialog text retained in the script. The majority of the pairs (33/40) fall below the red diagonal $x = y$ line.

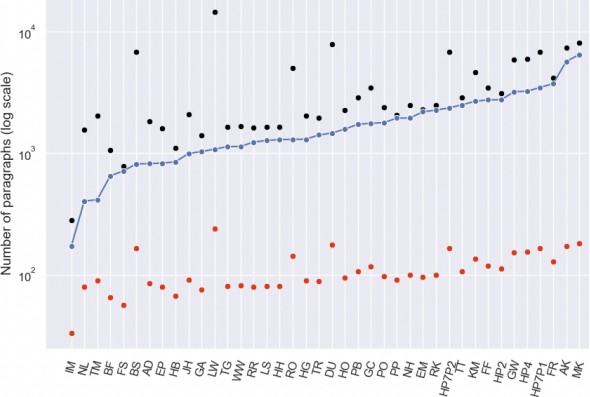

Figure 5: Books and scripts generally proceed in the same temporal order, so book-script alignments should increase monotonically. We plot the length of the longest increasing subsequence (LIS) of each $n$-part alignment in blue, and compare to the expected LIS (in red) for a random permutation ($2\sqrt{n}$). The black points denote the upper limit. The significantly higher LIS lengths prove that local Smith-Waterman alignments capture the global temporal order.

lated to be 1.09 using Cohen's d, signifying a large effect.

## 6.3 Sequential Orderings in Books and Adaptations

Narrative order in both books and films generally follow sequentially with time, despite the existence of literary devices such as flashbacks. We propose that the sequential order of events in books and scripts will largely coincide. Such a hypothesis cannot be tested with a global alignment algorithm such as *Book2Movie*, because such global alignments are forced to be sequential by design. But our **local** Smith-Waterman alignments permit a meaningful test, by quantifying the degree of monotonicity exhibited in a book-script alignment. SW aligns each book paragraph to an arbitrary paragraph in the script or let it remain unaligned.

To score aligment monotonicity, we compute the longest increasing subsequence (LIS) of the aligned book-script pair, representing the longest story portion in the book presented in the same order in the script. Figure 5 presents the length of LIS for all book-movie pairs in our dataset, along with the expected length according to chance. Logan and Shepp (1977) proved that the expected LIS of two random $n$-element permutations is $\sim 2\sqrt{n}$. All our alignments display much larger increasing subsequences, closer to the trivial upper bound of $n$.

## 6.4 A Modified Bechdel Test

The cultural importance of movies makes the unbiased representation of women in film an important issue. The Bechdel test (Bechdel, 1986) provides a simple framework for evaluating the representation of women in films. To pass the test, a film must satisfy three requirements: (i) There are two women in the film (ii) who talk with each other (iii) about something other than men. Previous works have focused on its usage for analysing films and society as a whole (Selisker, 2015) as well as automating the test (Agarwal et al., 2015). However, its binary nature fails to capture the degree of unbiased representation. Our ability to segment books into the respective portions which are/are not included in a corresponding movie script permits us to study the issues of adaptation through a Bechdel-inspired lens along a continuum.

Let $d$ represent the set of all dialogs in a book, $d_{f,b}$ denote the dialogs of female characters without any mention of male characters, and $rd$ encompass all retained dialogs in the film. We introduce the Bechdel representation ratio $B$ as the ratio between retention fraction of $d_{f,b}$ and overall retention fraction of dialogs.

$$B = \frac{|d_{f,b} \cap rd|}{|d_{f,b}|} \times \frac{|d|}{|rd|} \qquad (4)$$

| Book Name | Year (Book, Movie) | Retention % | Critic Opinion |
|---|---|---|---|
| The Lost World | 1995, 1997 | 13.8 | "*incredibly loose adaptation*" (Weiss, 2023) |
| Dune | 1965, 1984 | 14.1 | "*the last filmmaker you'd want to adapt your novel to the screen if you're looking for a faithful retelling* "(Davis, 2018) |
| The Grapes of Wrath | 1939, 1940 | 18.9 | - |
| Anna Karenina | 1878, 2012 | 22.1 | - |
| All the King's Men | 1946, 1949 | 26.4 | "*vary wildly from one another*" (Davis, 2018) |
| The Two Towers | 1954, 2002 | 79.8 | "*Surprisingly Book Accurate*" (Gass, 2019) |
| The Return of the King | 1955, 2003 | 81.2 | "*Surprisingly Book Accurate*" (Gass, 2019) |
| Harry Potter 2 | 1998, 2002 | 84.3 | "*followed the book the most closely*" (Collier, 2017) |
| Pride and Prejudice | 1813, 2005 | 89.4 | "*fairly faithful to the source*" (Blohm, 2016) |
| Emma | 1816, 1996 | 91.7 | "*adapted fairly faithfully to film*" (Blohm, 2016) |

Table 3: Faithfulness of screen adaptation: the five film scripts with the least/most book retention in our dataset.

We consider books with a Bechdel representation ratio $B > 1$ to be a good predictor of whether the film is likely to pass the Bechdel test. For a collection of 26 book-film pairs which have been assessed on the Bechdel test by crowdsourcing (Hickey, 2014), only 3 of the 10 films pass with $b \leq 1$, while 10 of 16 pass with $b > 1$, i.e. our algorithm correctly identifies the Bechdel test result for 17/26 film adaptations. More details about the data can be found in Appendix F.

## 7    Conclusion

In this paper, we have adapted a local dynamic programming alignment algorithm to efficiently align books and scripts and conducted a proof-of-concept analysis of 40 film adaptations, revealing insights into screenwriters' decision-making from various perspectives. Our work demonstrates that alignment methods can be used to identify which parts of books appear in movies, and opens up new research directions regarding how films are adapted from books. Future research directions include exploring patterns in screenwriting process that reflect the prevailing zeitgeist and cultural dynamics. By leveraging automated methods, it is possible to investigate how these patterns vary across different time periods, cultures as well as for characters of different genders and ethnicities.

One future study would be to create a model that can embed video frames and book text in the same embedding space essentially converting two different media to the same space that can create better

metrics to quantify similarity, yielding better alignments using the Smith-Waterman algorithm. This study can shed light onto how much the auditory and visual analysis strengthens the adaptation analysis process.

## Limitations

The *Removed Parts Annotation Set* discussed in Section 4 employs a non-standard segmentation unit called *book units*. While chapters and paragraphs are standard alternatives, they each face their own challenges. Chapters are too large to be annotated as fully removed or retained in film adaptation while annotating paragraphs will require a high number of human hours. Book units presented a promising alternative that required less resources and also created units more likely to be completely removed or retained in the film. However the manual annotator occasionally thought a book unit was partially retained in the film. For future research, annotating book paragraphs would be an alternative given sufficient resources, although our annotated book unit dataset remains a readily available option for evaluation.

The alignment of books with their film adaptations is inherently subjective. Different human annotators are likely to have disagreements in alignment construction. With ample resources, employing multiple human annotators and calculating inter-annotator agreement using Cohen's Kappa would provide a more accurate assessment. The inclusion of a third annotator to resolve conflicts would further enhance the quality of annotations and provide insights into the performance of different methods. However, due to limited resources, we did not pursue multiple annotations in this study.

Many of the books and film scripts cannot be distributed in its original form due to copyright issues. This creates a hurdle for future researchers. To address this, upon acceptance of this paper, we will share the code and a step-by-step replication strategy for reproducing the copyright-protected portion of the dataset. This will enable researchers with access to the book and movie script to reproduce the dataset and utilize our annotations.

We employ SBERT embeddings as the similarity metric, which have been trained on a dataset with texts of similar length to our task. Exploring the impact of training a specialized SBERT model on book and movie-specific data could provide further

insights. Computationally more expensive models such as SGPT (Muennighoff, 2022) may provide more accurate similarity measure.

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

# Appendix

## A  Manually Annotated Dataset

The six books that were manually annotated are listed in Table A.1. Two graduate students annotated the books. The only requirement for the annotators was that they had read and watched their respective book and film. They did not have any additional expertise related to literature or films. They were given the book units in a spreadsheet where they annotated the units with binary labels. In case of book units which they thought were partially retained, they were instructed beforehand to annotate it as retained if they thought the majority of the content was present in the film. Annotating a book took on average $\sim 4$ hours.

The HP2 film and book had 116 scenes and 18 chapters respectively. On the other hand, the HP4 film and book had 68 scenes and 37 chapters. Annotating alignment of chapter-scenes for a book-film pair took similar amount of time on average as retention annotation for a book.

## B  Complete Dataset

Information about all 40 book-films are listed in Table B.1. Each book-film pair is given a unique ID. The books are ordered lexicographically in the table using their IDs. Information about lengths of the books and the films are additionally provided in Table B.2.

## C  Illustrative Example Alignment

In Table C.1 and Table C.2, we show the alignment of a scene from the second Harry Potter film and the film *Room(2015)* with their respective original source novels. It is to be noted that the complete script and book had been aligned for both of them and we extract the particular scene's alignment to illustrate the following observations:

- In Table C.1, Script paragraph 429 is left unaligned correctly, showing the necessity of gapped alignments and the gap penalties as discussed in Section 3.1.

- Paragraph 430 and 431 are aligned with the same book paragraph, proving the necessity of many-to-many alignment instead of 1-1 alignment, also discussed in Section 3.1.

- In paragraph 438, the screenwriter assigns *Harry* the lines originally attributed to *Ron* in

the book, while also altering the text substantially. Despite this deviation from the original source material, our algorithm adeptly captures the fundamental core of the narrative, wherein a character laments their unequivocal failure to catch the train, and successfully aligns the pertinent textual components.

## D  Different Similarity Metrics

For a comparative analysis of SW+SBERT, we use 4 other similarity metrics both for our baseline greedy and SW alignment method. The metrics are described below:

**Jaccard:** The Jaccard index treats text as a bag-of-words and computes the similarity using the multiset of words present in both text segments:

$$J(a,b) = \frac{a \cap b}{a \cup b} \tag{5}$$

**TF-IDF:** The Term Frequency Inverse Document Frequency (TF-IDF) measures text similarity using the frequency of words weighted by the inverse of their presence in the texts, giving more weight to rare words. Here the set of documents is represented by the concatenated set of paragraphs from the book and film script.

**GloVe Mean Embedding**: We represent a text by the average of GloVe[3] embeddings of all the words in the text, using cosine similarity.

**Hamming Distance:** Given two paragraphs containing $n \geq m$ words, we decompose the longer paragraph into chunks of size $n/m$. The Hamming distance $h(.,.)$ is the fraction of chunks where chunk $i$ does not contain word $i$ from the shorter text. We then use $1 - h(.,.)$ as the similarity score.

## E  Faithfulness Labels for Film Adaptations

From different online reviews by critics, we compiled faithfulness labels for 18 film adaptations based on how faithful they were to their original novel. We only assigned labels when a critic review unambiguously described the film as either faithful or unfaithful. For example, for the film *Anna Karenina (2012)*, a review[4] describes the film as *"largely faithful"* but then goes on writing

---

[3] https://nlp.stanford.edu/projects/glove/
[4] https://creativeeccentric.wordpress.com/2013/01/21/

| ID | Book (Year) | Author | Film Adaptation (Year) | Film Length (Minutes) | No. of Book. Units | Human Annotation Retention% |
|---|---|---|---|---|---|---|
| HP2 | Harry Potter and The Chamber of Secrets | J. K. Rowling | Harry Potter and The Chamber of Secrets (2002) | 161 | 759 | 59.5 |
| HP4 | Harry Potter and The Goblet of Fire | J. K. Rowling | Harry Potter and The Goblet of Fire (2005) | 157 | 1400 | 27.6 |
| HP7P1 | Harry Potter and The Deathly Hallows | J. K. Rowling | Harry Potter and The Deathly Hallows Part 1 (2010) | 146 | 1605 | 40.3 |
| HP7P2 | Harry Potter and The Deathly Hallows | J. K. Rowling | Harry Potter and The Deathly Hallows Part 2 (2011) | 130 | 1605 | 24.7 |
| PP | Pride and Prejudice | Jane Austen | Pride and Prejudice (2005) | 127 | 498 | 48.6 |
| PB | The Princess Bride | William Goldman | The Princess Bride (1987) | 98 | 677 | 55.8 |

Table A.1: Data on the six book-script pairs whose books units are manually annotated as removed or retained. Additionally we create manual alignment of HP2 and HP4 book chapters with their movie scenes.

- *"The book is just short of 350,000 words so many choices had to be made."*

- *"effectively trims a significant portion of the book"*

- *"leaves out entire characters."*

- *"stays clear of Tolstoy's long discussions of Russian contemporary issues and the inner spiritual reflections of the characters."*

For such reviews, we opted not to create a label and denote it by the *?* symbol in Table E.1.

## F    Data for Bechdel Test

We combine three different sources to get the Bechdel test result for films in our dataset.

- For the film, *Dr. Jekyll and Mr. Hyde (1941)* we get the Bechdel test result from `https://h2g2.com/edited_entry/A87926584`.

- For the films *The Wonderful Wizard of Oz (2011)* and *M*A*S*H (1970)*, we use data from `https://lisashea.com/`.

- Hickey (2014) compiled a dataset[5] using the data from `https://bechdeltest.com` where Bechdel test results of films are stored by users. We intersected our dataset with theirs to get the result for the other 23 films.

The Bechdel test results, Bechdel, female and male representation ratios computed by our algorithm as discussed in Section 6.4 are presented in Table F.1.

---

[5]`https://github.com/fivethirtyeight/data/blob/master/bechdel/movies.csv`

| ID | Book Name | Book Year | Author | Film Name | Film Year | Genre |
|---|---|---|---|---|---|---|
| AD | Do Androids Dream of Electric Sheep | 1968 | Philip K Dick | Blade Runner | 1982 | Sci-fi |
| AK | Anna Karenina | 1878 | Leo Tolstoy | Anna Karenina | 2012 | Romance |
| BF | Big Fish: A Novel of Mythic Proportions | 1998 | Daniel Wallace | Big Fish | 2003 | Fantasy |
| BS | The Bourne Supremacy | 1986 | Robert Ludlum | The Bourne Supremacy | 2004 | Thriller |
| DU | Dune | 1965 | Frank Herbert | Dune | 1984 | Sci-fi |
| EM | Emma | 1885 | Jane Austen | Emma | 1996 | Romance |
| EP | The English Patient | 1992 | Michael Ondaatje | The English Patient | 1996 | Romance |
| FF | Far From The Madding Crowd | 1874 | Thomas Hardy' | Far from the Madding Crowd | 2015 | Romance |
| FR | The Fellowship of the Ring | 1954 | J. R. R. Tolkien | LOTR: The Fellowship of the Ring | 2001 | Fantasy |
| FS | Frankenstein; or, The Modern Prometheus | 1818 | Mary Shelley | Mary Shelley's Frankenstein | 1994 | Horror |
| GA | The Getaway | 1958 | Jim Thompson | The Getaway | 1972 | Thriller |
| GC | Get Carter 1 - Jack's Return Home | 1970 | Ted Lewis | Get Carter | 1971 | Thriller |
| GW | The Grapes of Wrath | 1939 | John Steinbeck | The Grapes of Wrath | 1940 | Drama |
| HB | The Hellbound Heart | 1986 | Clive Barker | Hellraiser | 1987 | Horror |
| HG | Hitchhiker's Guide to the Galaxy | 1979 | Douglas Adams | The Hitchhiker's Guide to the Galaxy | 2005 | Sci-fi |
| HH | The Haunting of Hill House | 1959 | Shirley Jackson | The Haunting | 1999 | Horror |
| HO | MASH: A Novel About Three Army Doctors | 1968 | Richard Hooker | M*A*S*H | 1970 | Comedy |
| HP2 | Harry Potter and the Chamber of Secrets | 1998 | JK Rowling | Harry Potter and the Chamber of Secrets | 2002 | Fantasy |
| HP4 | Harry Potter and the Goblet of Fire | 2000 | JK Rowling | Harry Potter and the Goblet of Fire | 2005 | Fantasy |
| HP7P1 | Harry Potter and the Deathly Hallows | 2007 | JK Rowling | Harry Potter and the Deathly Hallows | 2010 | Fantasy |
| HP7P2 | Harry Potter and the Deathly Hallows | 2007 | JK Rowling | Harry Potter and the Deathly Hallows | 2011 | Fantasy |
| IM | The Iron Man | 1968 | Ted Hughes | The Iron Giant | 1999 | Animation |
| JH | Strange Case of Dr Jekyll and Mr Hyde | 1886 | Robert Louis Stevenson | Dr. Jekyll and Mr. Hyde | 1941 | Sci-fi |
| KM | All the King's Men | 1946 | Robert Penn Warren | All the King's Men | 1949 | Drama |
| LS | I Know What You Did Last Summer | 1973 | Lois Duncan | I Know What You Did Last Summer | 1997 | Horror |
| LW | The Lost World | 1995 | Michael Crichton | The Lost World: Jurassic Park | 1997 | Sci-fi |
| MK | The Three Musketeers | 1844 | Alexandre Dumas | The Three Musketeers | 1993 | Action |
| NH | The Night of the Hunter | 1953 | Davis Grubb | The Night of the Hunter | 1955 | Drama |
| NL | Nothing Lasts Forever | 1979 | Roderick Thorp | Die Hard | 1988 | Thriller |
| PB | The Princess Bride | 1973 | William Goldman | The Princess Bride | 1987 | Comedy |
| PO | The Phantom of the Opera | 1910 | Gaston Leroux | The Phantom of the Opera | 2004 | Romance |
| PP | Pride and Prejudice | 1813 | Jane Austen | Pride and Prejudice | 2005 | Romance |
| RK | The Return of the King | 1955 | J. R. R. Tolkien | LOTR: The Return of the King | 2003 | Fantasy |
| RO | Room | 2010 | Emma Donoghue | Room | 2015 | Thriller |
| RR | Revolutionary Road | 1961 | Richard Yates | Revolutionary Road | 2008 | Romance |
| TG | The Grifters | 1963 | Jim Thompson | The Grifters | 1990 | Thriller |
| TM | The Time Machine | 1895 | H. G. Wells | The Time Machine | 1960 | Sci-fi |
| TR | The Road | 2006 | Cormac McCarthy | The Road | 2009 | Thriller |
| TT | The Two Towers | 1954 | J. R. R. Tolkien | LOTR: The Two Towers | 2002 | Fantasy |
| WW | The Wonderful Wizard of Oz | 1900 | Lyman Frank Baum | The Wizard of Oz | 1939 | Fantasy |

Table B.1: Complete dataset of 40 book and film adaptations with their IDs listed.

Table B.2: Length information of all book-films. A page is estimated to have 300 words.

| ID | Book Length (In Pages) | Script Length (In Pages) | Film Length (In Minutes) | No. of Script Paragraphs | No. of Book Paragraphs | Reten-tion% |
|---|---|---|---|---|---|---|
| AD | 243 | 79 | 117 | 1630 | 1818 | 61.3 |
| AK | 1264 | 93 | 130 | 2126 | 7402 | 22.1 |
| BF | 152 | 86 | 125 | 1926 | 1063 | 71.6 |
| BS | 851 | 75 | 108 | 1980 | 6784 | 27.3 |
| DU | 768 | 80 | 155 | 1516 | 7842 | 14.1 |
| EM | 575 | 49 | 124 | 1601 | 2285 | 91.7 |
| EP | 299 | 103 | 162 | 1725 | 1611 | 73.5 |
| FF | 496 | 72 | 119 | 2696 | 3471 | 68.7 |
| FR | 685 | 80 | 137 | 1708 | 4141 | 43.1 |
| FS | 261 | 100 | 123 | 1780 | 778 | 69.1 |
| GA | 187 | 83 | 122 | 1809 | 1403 | 65.4 |
| GC | 265 | 52 | 112 | 1340 | 3442 | 38.7 |
| GW | 762 | 86 | 129 | 1464 | 5884 | 18.9 |
| HB | 110 | 66 | 93 | 1784 | 1107 | 69.8 |
| HG | 175 | 81 | 110 | 1923 | 2023 | 56 |
| HH | 261 | 99 | 114 | 1798 | 1642 | 73 |
| HO | 228 | 100 | 116 | 1643 | 2255 | 47.4 |
| HP2 | 316 | 94 | 161 | 2278 | 3116 | 84.3 |
| HP4 | 706 | 44 | 157 | 2780 | 5994 | 34.8 |
| HP7P1 | 726 | 102 | 146 | 2196 | 6778 | 46.9 |
| HP7P2 | 726 | 105 | 130 | 2377 | 6778 | 33.8 |
| IM | 36 | 62 | 87 | 1438 | 279 | 70.2 |
| JH | 145 | 109 | 98 | 3998 | 2090 | 72.2 |
| KM | 809 | 75 | 110 | 1820 | 4616 | 26.4 |
| LS | 198 | 39 | 101 | 848 | 1644 | 40.2 |
| LW | 467 | 108 | 129 | 2719 | 14456 | 13.8 |
| MK | 837 | 93 | 105 | 3378 | 8132 | 29.3 |
| NH | 267 | 80 | 92 | 2217 | 2483 | 68 |
| NL | 234 | 88 | 131 | 1887 | 1567 | 74.9 |
| PB | 332 | 74 | 98 | 2028 | 2876 | 56.1 |
| PO | 304 | 118 | 143 | 1723 | 2371 | 60.6 |
| PP | 443 | 85 | 127 | 1662 | 2065 | 89.4 |
| RK | 493 | 100 | 201 | 2344 | 2492 | 81.2 |
| RO | 361 | 75 | 118 | 2273 | 5036 | 31 |
| RR | 384 | 57 | 119 | 1436 | 1614 | 64 |
| TG | 182 | 75 | 110 | 1556 | 1650 | 63.4 |
| TM | 116 | 80 | 96 | 1408 | 2021 | 51.1 |
| TR | 214 | 83 | 111 | 1403 | 1957 | 62.3 |
| TT | 570 | 128 | 179 | 3834 | 2876 | 79.8 |
| WW | 204 | 108 | 101 | 2061 | 1665 | 72.9 |

Table C.1: Alignment of the scene where Harry and Ron misses the train to Hogwarts in HP2.

| Script Paragraph # | Script Paragraph | Aligned Book Paragraph |
|---|---|---|
| 428 | MRS. WEASLEY (O.S.) Come, Ginny. We'll get you a seat. | I'll take Ginny and you two come right after us ,' Mrs Weasley told Harry and Ron, grabbing Ginny's hand and setting off . In the blink of an eye they were gone. |
| 429 | 32B INT. KING'S CROSS 32B | |
| 430 | Ron glances at the clock. | Let's go together, we've only got a minute ,' Ron said to Harry. Let's go together, we've only got a minute ,' Ron said to Harry. |
| 431 | RON We better hurry. | |
| 432 | Harry nods, leans into his trolley and – CRASH! – hits the barrier and bounces back into Ron. A GUARD glowers. | Harry made sure that Hedwig's cage was safely wedged on top of his trunk and wheeled his trolley about to face the barrier . He felt perfectly confident; this wasn't nearly as uncomfortable as using Floo powder . Both of them bent low over the handles of their trolleys and walked purposefully towards the barrier, gathering speed . A few feet away from it, they broke into a run and CRASH . ASH. |
| 433 | GUARD What in blazes d'you two think you're doing? | Both trolleys hit the barrier and bounced backwards . Ron's trunk fell off with a loud thump, Harry was knocked off his feet, and Hedwig's cage bounced onto the shiny floor and she rolled away, shrieking indignantly . People all around them stared and a guard nearby yelled ,' What in blazes d'you think you're doing? ' |
| 434 | HARRY Sorry. Lost control of the trolley. | Lost control of the trolley ,' Harry gasped, clutching his ribs as he got up . Ron ran to pick up Hedwig, who was causing such a scene that there was a lot of muttering about cruelty to animals from the surrounding crowd. |
| 435 | (to Ron) Why can't we get through? | Why can't we get through?' Harry hissed to Ron. |
| 436 | RON I dunno. The gateway's sealed itself for some reason. | We're going to miss the train ,' Ron whispered .' I don't understand why the gateway's sealed itself ...' Harry looked up at the giant clock with a sickening feeling in the pit of his stomach. |
| 437 | As Ron presses his ear to the barrier, the CLOCK CHIMES. | Ten seconds...nine seconds...He wheeled his trolley forward cautiously until it was right against the barrier, and pushed with all his might. |
| 438 | HARRY The train leaves at exactly eleven o'clock. We've missed it. | Three seconds...two seconds...one second ...' It's gone ,' said Ron, sounding stunned. |
| 439 | RON Can't hear a thing. | The Dursleys haven't given me pocket money for about six years .' Ron pressed his ear to the cold barrier . ' |
| 440 | (a sudden thought) Harry. If we can't get through, maybe Mum and Dad can't get back. | What're we going to do? I don't know how long it'll take Mum and Dad to get back to us .' They looked around. |
| 441 | HARRY Maybe we should go wait by the car. | I think we'd better go and wait by the car ,' said Harry . We're attracting too much atten -' ' Harry! |
| 442 | RON The car! | said Ron, his eyes gleaming .' The car!' ' What about it?' ' We can fly the car to Hogwarts! |

Table C.2: Aligning the scene depicting *Jack* feigning death in front of his captor, *Old Nick*, in the film *Room (2015)* with its source book.

| Script Para-graph # | Script Paragraph | Book Paragraph |
|---|---|---|
| 833 | The door beeps, and Ma rolls Jack up, very fast. We're in the dark with him, the rug pressing close. | Beep beep. |
| 834 | Sound of Old Nick stepping in. Proud of himself. | " Here you go . " That's Old Nick's voice . He sounds like always . He doesn't even know what's happened about me dying . " Antibiotics, only just past the sell - by . For a kid you break them in half, the guy said . " Ma doesn't answer . " " Here you go . " That's Old Nick's voice . He sounds like always . He doesn't even know what's happened about me dying . " Antibiotics, only just past the sell - by . For a kid you break them in half, the guy said . " Ma doesn't answer . " |
| 835 | OLD NICK (O.S.) Antibiotics. | |
| 836 | The door shuts with a boom. | |
| 837 | OLD NICK (O.S.) (CONT'D) Are you nuts, wrapping a sick kid up in that? | |
| 838 | Ma's voice is very close because she's bent over the rug. | Is he in the rug? |
| 839 | MA (O.S.) He got worse in the night. | He got worse in the night and this morning he wouldn't wake up . " Nothing. |
| 840 | She waits, giving Old Nick time to figure it out. | Then Old Nick makes a funny sound . " Are you sure? " |
| 841 | In the rug, Jack lies frozen except for his flickering eyes. | " Am I sure? " Ma shrieks it, but I don't move, I don't move, I'm all stiff no hearing no seeing no nothing. |

| ID | Movie | Faithfulness Source | Faithful | Retention% |
|---|---|---|---|---|
| LW | The Lost World: Jurassic Park (1997) | https://www.simplemost.com/movie-adaptations-that-were-nothing-like-the-book/ | × | 13.8 |
| DU | Dune (1984) | https://www.simplemost.com/movie-adaptations-that-were-nothing-like-the-book/ | × | 14.1 |
| AK | Anna Karenina (2012) | https://creativeeccentric.wordpress.com/2013/01/21/ | ? | 22.1 |
| KM | All the King's Men (1949) | https://www.simplemost.com/movie-adaptations-that-were-nothing-like-the-book/ | × | 26.4 |
| BS | The Bourne Supremacy (2004) | https://www.simplemost.com/movie-adaptations-that-were-nothing-like-the-book/ | × | 27.3 |
| MK | The Three Musketeers (1993) | https://www.nytimes.com/1993/11/12/movies/reviews-film-once-more-into-the-fray-for-athos-porthos-et-al.html | × | 29.3 |
| HP7P2 | Harry Potter 7 Part 2 (2011) | https://screenrant.com/harry-potter-most-accurate-movies-based-books/ | ? | 33.8 |
| HP4 | Harry Potter 4 (2005) | https://screenrant.com/harry-potter-most-accurate-movies-based-books/ | ? | 34.8 |
| LS | I Know What You Did Last Summer (1997) | https://www.kentuckypress.com/9780813136554/the-philosophy-of-horror/ | × | 40.2 |
| FR | LOTR: The Fellowship of the Ring (2001) | https://screenrant.com/movies-surprisingly-book-accurate/#the-lord-of-the-rings | ✓ | 43.1 |
| HP7P1 | Harry Potter 7 Part 1 (2010) | https://screenrant.com/harry-potter-most-accurate-movies-based-books | ? | 46.9 |
| AD | Blade Runner (1982) | https://www.simplemost.com/movie-adaptations-that-were-nothing-like-the-book/ | × | 61.3 |
| TG | The Grifters (1990) | https://www.cbr.com/movies-like-original-book/ | ✓ | 63.4 |
| FS | Mary Shelley's Frankenstein (1994) | https://collider.com/mary-shelleys-frankenstein-adaptation/ | ✓ | 69.1 |
| IM | The Iron Giant (1999) | https://www.indiewire.com/features/general/5-things-you-might-not-know-about-brad-birds-the-iron-giant-107519/ | × | 70.2 |
| WW | The Wizard of Oz (1939) | https://collider.com/book-to-film-adaptations-not-like-the-books/ | × | 72.9 |
| NL | Die Hard (1988) | https://collider.com/movies-nothing-like-book | × | 74.9 |
| TT | LOTR: The Two Towers (2002) | https://screenrant.com/movies-surprisingly-book-accurate/ | ✓ | 79.8 |
| RK | LOTR: The Return of the King (2003) | https://screenrant.com/movies-surprisingly-book-accurate/ | ✓ | 81.2 |
| HP2 | Harry Potter 2 (2002) | https://www.theodysseyonline.com/5-movies-based-on-books-that-are-actually-like-the-book | ✓ | 84.3 |
| PP | Pride and Prejudice (2005) | https://interestingliterature.com/2016/03/jane-austen-adaptations-throughout-history/ | ✓ | 89.4 |
| EM | Emma (1996) | https://interestingliterature.com/2016/03/jane-austen-adaptations-throughout-history/ | ✓ | 91.7 |

Table E.1: Faithfulness of 18 film adaptations have 0.65 Spearman correlation with retention percentage computed by SW+SBERT. For four film adaptations, the reviews were not deemed to be convincing enough to label the films as faithful or unfaithful, denoted by the *?* symbol in the Faithful column.

Table F.1: A book/script with a Bechdel ratio $b > 1$ is a good predictor of the offical Bechdel test, correctly predicting 17/26 films.

| ID | Book/Film Name | Female Represen-tation Ratio | Male Representa-tion Ratio | Bechdel Representa-tion Ratio | Bechdel Test | SW+SBERT Prediction Correct-ness |
|---|---|---|---|---|---|---|
| HO | Mash | 0.767 | 1.052 | 0.711 | Pass | ✗ |
| PB | The Princess Bride | 0.805 | 1.036 | 0.85 | Fail | ✓ |
| HP7P2 | Harry Potter 7 Part 2 | 0.822 | 1.065 | 0.852 | Fail | ✓ |
| BF | Big Fish | 0.93 | 1.015 | 0.913 | Fail | ✓ |
| KM | All the King's Men | 0.951 | 1.037 | 0.95 | Fail | ✓ |
| NL | Nothing Lasts Forever | 0.963 | 1.023 | 0.902 | Fail | ✓ |
| HP4 | Harry Potter 4 | 0.963 | 1.011 | 0.93 | Fail | ✓ |
| FR | The Fellowship of the Ring | 0.998 | 1 | 0.992 | Fail | ✓ |
| PP | Pride and Prejudice | 1 | 1 | 0.998 | Pass | ✗ |
| EM | Emma | 1.001 | 0.998 | 1.001 | Pass | ✓ |
| RR | Revolutionary Road | 1.011 | 0.993 | 1.003 | Pass | ✓ |
| LS | I Know What You Did Last Summer | 1.013 | 0.973 | 1.016 | Pass | ✓ |
| WW | The Wonderful Wizard of Oz | 1.017 | 0.969 | 1.011 | Pass | ✓ |
| HH | The Haunting of Hill House | 1.017 | 0.967 | 1.02 | Pass | ✓ |
| AK | Anna Karenina | 1.02 | 0.987 | 1.026 | Fail | ✗ |
| RK | The Return of the King | 1.035 | 0.997 | 1.03 | Fail | ✗ |
| FS | Frankenstein | 1.035 | 0.984 | 1.034 | Fail | ✗ |
| FF | Far From The Madding Crowd | 1.048 | 0.967 | 1.045 | Pass | ✓ |
| HB | The Hellbound Heart | 1.064 | 0.925 | 1.048 | Pass | ✓ |
| TG | The Grifters | 1.08 | 0.952 | 1.08 | Pass | ✓ |
| AD | Androids Dream of Electric Sheep | 1.084 | 0.974 | 1.071 | Fail | ✗ |
| GC | Get Carter (Jack's Return Home) | 1.116 | 0.952 | 1.078 | Fail | ✗ |
| PO | The Phantom of the Opera | 1.116 | 0.953 | 1.113 | Fail | ✗ |
| HP7P1 | Harry Potter 7 Part 1 | 1.117 | 0.957 | 1.068 | Pass | ✓ |
| JH | Dr. Jekyll and Mr. Hyde | 1.14 | 0.994 | 1.13 | Pass | ✓ |