# OpenReview forum: "Analyzing Film Adaptation through Narrative Alignment"
_EMNLP/2023/Conference — EMNLP 2023 Main_

### Official Review · Reviewer_CXES · 2023-08-05

**Soundness:** 4

**Excitement:**

4: Strong: This paper deepens the understanding of some phenomenon or lowers the barriers to an existing research direction.

**Missing References:**

Scene reordering in movie script alignment. (Lambert et al., 2013) is an older relevant work. Stephen Follows' blog contains useful studies of screenplay vs film runtime.

**Paper Topic And Main Contributions:**

This paper offers an analysis of adaptations of novels into films. It provides three datasets, two of them with manual annotations. It tests techniques to align novels and films that is an improvement compared to previous work (finding that local alignment with SBERT performs best). It also explores ways to measure faithfulness to the source material. Last, they provide a measure w.r.t. the extent of retention of dialogs that pass the Bechdel Test in each adaptation.

**Questions For The Authors:**

What motivated your choices re: algorithms and models? e.g. why SBERT vs SGPT? why GloVE vs ELMo?

**Reasons To Accept:**

The study of adaptations is an underexplored and underresourced task in narrative NLP. This paper could kickstart more work in this space, particularly due to its datasets and its detailed experimental comparisons for alignment and faithulness.

It seems the authors were in a tough spot between ethics (respecting copyright) and reproducibility (providing the dataset). Their planned solution to provide a step-by-step replication strategy seems to be a reasonable compromise.

**Reasons To Reject:**

The chosen techniques used for alignment (e.g. Smith-Waterman+ SBERT) and faithfulness (GloVE, Hamming distance etc) may be somewhat dated and it's possible that newer / more tailored techniques might perform better (e.g. newer or finetuned sentence representations; contextualized embeddings; algorithms used in the detection of paraphrases or plagiarism).*

The provided datasets are somewhat small, especially gold alignments, and may be insufficient to judge the effectiveness of the chosen techniques. However, it is understandable that copyright protections and limited resources posed constraints.

\* EDIT: However, as the authors explained, they deliberately gave preference to techniques that would not be very computationally expensive.

**Reproducibility:**

3: Could reproduce the results with some difficulty. The settings of parameters are underspecified or subjectively determined; the training/evaluation data are not widely available.

**Reviewer Confidence:**

4: Quite sure. I tried to check the important points carefully. It's unlikely, though conceivable, that I missed something that should affect my ratings.

**Typos Grammar Style And Presentation Improvements:**

Suggestion for the code - it would be good to include package versions in your requirements file.

---

> ### Author Rebuttal · Authors · 2023-08-29
>
> Thank you for your detailed review.  We will incorporate your new references into the final version of our paper.
>
> Our primary technical goal in this paper was to test how similarity metrics from different domains (such as sentence embedding based, mean word embedding based, and traditionally popular metrics such as Jaccard, TF-IDF) performed for our particular use-case of quantifying similarity between book paragraphs and script paragraphs. We mostly opted to use the lightweight options unless the lightweight option proved much weaker compared to its alternative. The definition of "much" here is subjective.
>
> We preferred SBERT over SGPT for the following reasons:
> 1. The reported performance of SGPT using a similar number of parameters as SBERT is typically similar or poorer in most datasets—SGPT only became better when the number of parameters became multiple magnitudes higher. But this also creates larger computational resource requirements, which are challenging in our research environment.
> 2. SBERT has a better/simpler API to use and has been more widely used in the research community than SGPT, admittedly because it got published four years earlier.
> 3. As we discuss in Section 3.3, SBERT trained using MS-Marco dataset seemed to align with our use-case with regards to the lengths of texts we compute the similarity metric for, one short text against a longer text.
>
> Nevertheless, we agree with the reviewer that an analysis between SBERT and SGPT will be interesting for our experiments. We will try to do a subset of our experiments using two SGPT models, one model with similar number parameters as SBERT and the other being the largest SGPT model and add a comparative analysis in the camera-ready version upon acceptance of the paper.
>
> The mean word pooling method we used helped more in setting a baseline performance level as opposed to increasing the performance. Elmo usually uses a higher embedding dimension compared to GloVe. We opted for the lightweight GloVe here for resource demands.

---

### Official Review · Reviewer_ybnJ · 2023-08-06

**Soundness:** 4

**Excitement:**

4: Strong: This paper deepens the understanding of some phenomenon or lowers the barriers to an existing research direction.

**Missing References:**

There has also been some work on aligning movie scripts with subtitles, see Lison, P., & Meena, R. (2016, December). Automatic turn segmentation for movie & tv subtitles. In 2016 IEEE Spoken Language Technology Workshop (SLT) (pp. 245-252). IEEE.

**Paper Topic And Main Contributions:**

The paper presents an approach to automatically analyse film adaptation of books by running a local alignment algorithm between the book and the corresponding movie script. The approach is tested on 40 adaptations and relies on the Smith-Waterman algorithm together with similarity measures based on sentence-BERT. Those alignments are then used to investigate various aspects of the film adaptation, such as the faithfulness of the adaptation or the preservation of the narrative order.

**Questions For The Authors:**

- If I understood correctly, the text segmentation relies on "book units" for the text of the book, and "scenes" for the text of the script.  What is the typical length of a scene? Have you experimented with other segmentation choices for the script than scenes?
- did you investigate whether the relative position of a book unit in a book had an influence on the retention probability? For instance, it would be interesting to see whether paragraphs occurring in the first chapter (often providing some general introduction to the plot) were less likely to be retained.

**Reasons To Accept:**

- the paper is clear and well-written
- the experimental design is generally sound
- the paper investigates an interesting question and provides interesting empirical data on film adaptation

**Reasons To Reject:**

- the question being investigated is rather niche, and might not be of interest for a broad audience.
- the discussion of related work does not mention at any point the large body of work on alignment algorithms developed for machine translation (to align sentences or paragraphs in parallel or comparable corpora).

**Reproducibility:**

4: Could mostly reproduce the results, but there may be some variation because of sample variance or minor variations in their interpretation of the protocol or method.

**Reviewer Confidence:**

3: Pretty sure, but there's a chance I missed something. Although I have a good feel for this area in general, I did not carefully check the paper's details, e.g., the math, experimental design, or novelty.

**Typos Grammar Style And Presentation Improvements:**

- the acronyms "PB" and "PP" are only explained in the appendix.

---

> ### Author Rebuttal · Authors · 2023-08-29
>
> Thank you for your detailed review.
>
> We agree that the question being investigated has historically been niche, but with the advent of GPT-3 and more advanced models, this has coincidentally come into limelight. AI systems are increasingly being proposed to automate creative tasks in the entertainment industry. The Writers Guild of America has been on strike since May regarding multiple issues, one of them being “Regulate use of artificial intelligence on MBAcovered projects: AI can’t write or rewrite literary material; can’t be used as source material; and MBA-covered material can’t be used to train AI” [from “WGA Negotiations—Status as of May 1, 2023”]. We believe our quantitative analysis tools will make the screenwriting adaptation process an increasingly important area of study. Potential application of our tool for both sides of the debate may include:
> 1. Creating training dataset for generative AI for rewriting creative materials into different formats.
> 2. Analyzing the differences between materials created by humans and AI, as well as identifying areas where AI may have limitations.
>
> We agree that we should refer to the alignment methods developed for statistical (Lopez, 2008) and neural machine translation such as  Bahdanau et al., 2014 and discuss how the lengths of texts in our dataset make it challenging to use them out-of-box due to resource scarcity and memory requirements. We would also add discussion on how monolingual parallel corpus are created using alignment algorithms (Paun, 2021). Upon acceptance of the paper,  we would also add the missing references related to movie scripts in the camera-ready version and we thank the reviewer for pointing them out.
>
>
> Our book unit segmentation and scene segmentation makes the dataset annotation and evaluation process simpler, more intuitive and more interpretable. The typical length of a scene is ~28 paragraphs where each script paragraph represents either a setup of the scene or a single dialog. We did not experiment with other segmentation choices for scripts, because scenes seemed to be the most intuitive option.
>
> We did perform some preliminary experiments where we divided the book into three parts: setup (first 25%), confrontation (next 50%), and resolution (last 25%).  But we did not find any significant correlation between the three parts with the retention probability over our dataset of 40 book-movie pairs. But it is possible that some smaller specific parts of the books exhibit a pattern of being removed or retained majority of the time. Figure 3 does show us that the portions at the very beginning of books tend to be disproportionately removed, which somewhat supports the reviewer's hypothesis. We will further investigate whether the relative position of the book units have any correlation with the probability of retention, and report it in the camera ready version of the paper.

---

### Official Review · Reviewer_MAg3 · 2023-08-11

**Soundness:** 4

**Excitement:**

4: Strong: This paper deepens the understanding of some phenomenon or lowers the barriers to an existing research direction.

**Paper Topic And Main Contributions:**

The paper analyzes the alignment of literary works with their corresponding film adaptation. An alignment technique based on Smith-Waterman's sequence alignment algorithm and NLP techniques for computing text similarity is introduced. The paper further introduces three manually annotated datasets to enhance research endeavors within this domain. The paper's main contributions include a deep evaluation of the proposed alignment method, insights into adaptation patterns such as retention of book parts, dialog representation, sequential orderings, and gender representation. The paper's findings have practical implications in advancing computationally aided linguistic analysis and understanding of the adaptation process.

**Questions For The Authors:**

A)	Given the limitations of a text-centric approach, have you explored how incorporating visual or auditory analysis to capture essential cinematic elements could enhance the quality of interpreting book-film adaptations?

**Reasons To Accept:**

This paper presents a compelling approach to the analysis of book-film adaptation, introducing an alignment method that demonstrates superior performance compared to robust baselines. The thorough evaluation of this algorithm and the incisive investigation into adaptation patterns yield valuable insights that enrich both scholarly and practical perspectives within the realms of screenwriting and adaptation. The paper stands to offer the NLP community an improved methodology and curated datasets for deeper understanding and exploration of the complex interplay between textual sources and cinematic adaptations.

**Reasons To Reject:**

The paper offers a valuable approach to analyzing book-film adaptation but has some weaknesses to consider. It primarily focuses on textual analysis, potentially overlooking important cinematic elements like aesthetics, sound, and performance. This limitation could restrict the comprehensive understanding of the adaptation process and potentially curtail the depth of insights attainable. Moreover, the heavy reliance on textual similarity measures may oversimplify the intricate process of film adaptation, posing a risk of misconstrued adaptation quality. Such simplification might disregard the intricate interplay of elements like narrative tempo, emotional subtleties, and creative choices that profoundly influence the quality of film adaptations.

**Reproducibility:**

4: Could mostly reproduce the results, but there may be some variation because of sample variance or minor variations in their interpretation of the protocol or method.

**Reviewer Confidence:**

5: Positive that my evaluation is correct. I read the paper very carefully and I am very familiar with related work.

**Typos Grammar Style And Presentation Improvements:**

At line 325 and 825, the phrase "In case" should be corrected to "In the case".
At line 379, the typographical error "unitsin" should be corrected to read as "units in".

---

> ### Author Rebuttal · Authors · 2023-08-28
>
> Thank you for your detailed review.
>
> We agree that visual and auditory analysis can enhance the quality of book-to-film alignment, and script analysis in general. We explored how the text-to-text alignment methods can capture how screenwriters are adapting the book writers' books into film scripts. Our current method draws a parallel between book writers and screenwriters. But we agree that a film goes through more modifications as the director, editor, actors and others bring in their own ideas which can only be captured through audio+video analysis. One future study would be to create a model that can embed video frames and book text in the same embedding space essentially converting two different media to the same space. Then we could use embedding similarity to quantify our similarity metric and use it with the Smith-Waterman alignment algorithm. Embedding video frames and related book text in the same space is an interesting research problem itself.  We will propose such a project in the future work section of our final version of this paper.
>
> Nevertheless, the first step in the book-to-film adaptation process is always the screenwriter creating the script, which arguably contains the biggest decisions to be made in the adaptation process. We successfully showed how alignment between the book and the script alone can reveal significant insights into the adaptation process. A future work should definitely dive deeper into how much the auditory and visual analysis strengthen the adaptation analysis process.

---

### Meta-Review · Area_Chair_yNQq · 2023-09-15

**Recommendation:** 5

**Metareview:**

There is a perfect agreement among reviewers about the soundness and excitement of this paper. Some reasons to reject do not have to do with soundness or excitement about the paper (research question is too niche, neglect of cinematic elements, literature on alignment, …), but may be explored in future work.

Paper Topic And Main Contributions:

This paper introduces a novel approach to analyzing the alignment between literary works and their film adaptations by employing the Smith-Waterman sequence alignment algorithm and sentence-BERT-based similarity measures. It presents three datasets, including two with manual annotations, and evaluates the alignment technique across 40 adaptations. The study delves into various aspects of the adaptation process, shedding light on patterns like narrative faithfulness, retention of book elements, dialog representation, and gender portrayal. These findings hold significant practical implications for advancing computational linguistic analysis and enhancing our understanding of the adaptation process, making it a valuable contribution to the field.

Reasons To Accept:

* Novelty
    * The paper addresses an underexplored area in narrative NLP, potentially catalyzing more research in this domain due to its valuable datasets and comprehensive experimental comparisons for alignment and faithfulness.
    * The inclusion of curated datasets enriches the research community's resources for further exploration of the intricate relationship between textual sources and cinematic adaptations.
    * The proposed alignment method demonstrates superior performance compared to established baselines, indicating the paper's potential to advance the field of NLP.
* Quality of the Methodology
    * The paper investigates a compelling question regarding book-film adaptation and offers interesting empirical data, contributing valuable insights to both scholarly and practical discussions in screenwriting and adaptation studies.
    * The authors have navigated the ethical challenge of copyright while ensuring reproducibility by providing a well-considered step-by-step replication strategy.
    * The experimental design is robust, enhancing the credibility of the presented results.
* Quality of the writing
    * The paper is well-written and clear, making it accessible to a broad audience.

Reasons To Reject:

* The heavy reliance on textual similarity measures oversimplifies the complex film adaptation process, potentially leading to misconstrued assessments of adaptation quality, as it doesn't consider factors like narrative tempo, emotional subtleties, and creative choices that significantly influence film adaptations.
* The chosen techniques for alignment and faithfulness assessment (e.g., Smith-Waterman, SBERT, GloVE, Hamming distance) may be outdated, and more recent or tailored methods might yield better results. This suggests a potential need for a more up-to-date approach to ensure the paper's findings remain relevant.
* The provided datasets, especially the gold alignments, are relatively small, which may limit the assessment of the effectiveness of the chosen techniques. While copyright limitations and resource constraints are acknowledged, larger datasets could provide more robust and reliable results.

---

### Decision · Program_Chairs · 2023-10-07

**Decision:**

Accept-Main

**Comment:**

There is a perfect agreement among reviewers about the soundness and excitement of this paper. Some reasons to reject do not have to do with soundness or excitement about the paper (research question is too niche, neglect of cinematic elements, literature on alignment, …), but may be explored in future work.

Paper Topic And Main Contributions:

This paper introduces a novel approach to analyzing the alignment between literary works and their film adaptations by employing the Smith-Waterman sequence alignment algorithm and sentence-BERT-based similarity measures. It presents three datasets, including two with manual annotations, and evaluates the alignment technique across 40 adaptations. The study delves into various aspects of the adaptation process, shedding light on patterns like narrative faithfulness, retention of book elements, dialog representation, and gender portrayal. These findings hold significant practical implications for advancing computational linguistic analysis and enhancing our understanding of the adaptation process, making it a valuable contribution to the field.

Reasons To Accept:

* Novelty
    * The paper addresses an underexplored area in narrative NLP, potentially catalyzing more research in this domain due to its valuable datasets and comprehensive experimental comparisons for alignment and faithfulness.
    * The inclusion of curated datasets enriches the research community's resources for further exploration of the intricate relationship between textual sources and cinematic adaptations.
    * The proposed alignment method demonstrates superior performance compared to established baselines, indicating the paper's potential to advance the field of NLP.
* Quality of the Methodology
    * The paper investigates a compelling question regarding book-film adaptation and offers interesting empirical data, contributing valuable insights to both scholarly and practical discussions in screenwriting and adaptation studies.
    * The authors have navigated the ethical challenge of copyright while ensuring reproducibility by providing a well-considered step-by-step replication strategy.
    * The experimental design is robust, enhancing the credibility of the presented results.
* Quality of the writing
    * The paper is well-written and clear, making it accessible to a broad audience.

Reasons To Reject:

* The heavy reliance on textual similarity measures oversimplifies the complex film adaptation process, potentially leading to misconstrued assessments of adaptation quality, as it doesn't consider factors like narrative tempo, emotional subtleties, and creative choices that significantly influence film adaptations.
* The chosen techniques for alignment and faithfulness assessment (e.g., Smith-Waterman, SBERT, GloVE, Hamming distance) may be outdated, and more recent or tailored methods might yield better results. This suggests a potential need for a more up-to-date approach to ensure the paper's findings remain relevant.
* The provided datasets, especially the gold alignments, are relatively small, which may limit the assessment of the effectiveness of the chosen techniques. While copyright limitations and resource constraints are acknowledged, larger datasets could provide more robust and reliable results.